# Children’s Exposure to Television Food Advertising Contributes to Strong Brand Attachments

**DOI:** 10.3390/ijerph16132358

**Published:** 2019-07-03

**Authors:** Bridget Kelly, Emma Boyland, Lesley King, Adrian Bauman, Kathy Chapman, Clare Hughes

**Affiliations:** 1Early Start, School of Health and Society, University of Wollongong, NSW 2522, Australia; 2Department of Psychological Sciences, University of Liverpool, Liverpool L3 5TR, UK; 3Prevention Research Collaboration, University of Sydney, Sydney, NSW 2006, Australia; 4School of Life and Environment Sciences, Faculty of Science, University of Sydney, Sydney, NSW 2006, Australia; 5School of Medicine & Public Health, University of Newcastle, Callaghan, NSW 2308, Australia; 6Cancer Council NSW, Sydney, NSW 2011, Australia

**Keywords:** food, beverage, advertising, television, marketing, child, brand

## Abstract

Children’s exposure to unhealthy food marketing is one factor contributing to childhood obesity. The impact of marketing on children’s weight likely occurs via a cascade pathway, through influences on children’s food brand awareness, emotional responses, purchasing and consumption. Thus, building emotional attachments to brands is a major marketing imperative. This study explored Australian children’s emotional attachments to food and drink brands and compared the strength of these attachments to their food marketing exposure, using television viewing as a proxy indicator. A cross-sectional face-to-face survey was conducted with 282 Australian children (8–12 years). Children were asked to indicate their agreement/disagreement with statements about their favourite food and drink brands, as an indicator of the strength and prominence of their brand attachments. Questions captured information about minutes/day of television viewing and the extent that they were exposed to advertising (watched live or did not skip through ads on recorded television). For those children who were exposed to advertisements, their age and commercial television viewing time had significant effects on food and drink brand attachments (*p* = 0.001). The development of brand attachments is an intermediary pathway through which marketing operates on behavioural and health outcomes. Reducing children’s exposure to unhealthy food marketing should be a policy priority for governments towards obesity and non-communicable disease prevention.

## 1. Introduction

Children’s exposure to marketing for unhealthy foods and beverages is widely accepted by leading international health organisations to be a causal factor contributing to childhood overweight and obesity. The World Health Organization’s (WHO) Commission on Ending Childhood Obesity confirmed that the evidence on the impact of unhealthy food marketing on childhood obesity is unequivocal [1]. Reducing children’s exposure to this marketing is a global priority for obesity and non-communicable disease prevention [1,2,3].

The effects of marketing exposure on children’s diet and health outcomes are likely to occur via a cascade pathway, such that repeated exposures lead to food brand recognition, brand affect or emotional responses, which in turn lead to food purchase and, finally, consumption behaviours [4]. Children and adolescents are a major target market for the food and beverage industry, given their own spending power, influence over household purchases and their potential as adult consumers [5]. Marketing campaigns seek to influence intermediate affective responses to ensure long term brand loyalty and also stimulate extant product sales [6].

The term “branding” refers to a unique name and/or symbol (e.g., logo) that identifies a product and distinguishes it from that of competitors [7]. The concept of brand is a dominant feature of marketing, where a brand is perceived to be a “living entity with a personality with which we can form a relationship and that can evolve over time” [8]. Brands become an anchor through which beliefs about brand attributes and benefits, and perceptions about users of the brand, are made; referred to as “brand image” [9]. This brand image is curated through marketing campaigns, which tailor the image of the brand to suit the needs and wants of the target market. Successful marketing leads to a set of positive associations with the brand, creating a powerful and lasting affinity with, or loyalty to, the brand [10]. The effect of brand image has been found to operate in children, whereby, among 8–12-year olds, agreement with hedonic brand attributes (e.g., “It is cheerful/fun”) has been shown to predict purchase intent more than utility attributes (e.g., “It is useful”) [11].

The development of specific and desirable brand images is crucial to products gaining market share. The promotion of products based solely on their real attributes is rarely useful in terms of encouraging brand switching or uptake. Within categories of processed packaged foods, in the absence of branding, products may be ostensibly interchangeable, such as Coca-Cola and Pepsi. Branding of these otherwise similar products has been found to dramatically influence expressed behavioural preferences based on measured brain responses [12].

Yet branding extends further than merely developing associations with a particular brand. It also involves the transfer of brand qualities into individuals’ own personae [13]. This may manifest in the wearing of branded clothing, the adoption of brand characteristics into youths’ self-identity, and to the evaluation of peers based on their branding choices [14]. Brands are used as symbols of social identity, including social class and status, and individual qualities [15]. Such brand attachments are an even stronger predictor of actual consumer behaviour than brand attitudes [16]. Attachment involves a cognitive and emotional connection or bond between a person and the brand, referred to as brand–self connection [17]. Brand–self connection forms when consumers hold brand associations that they can relate to their selves, including associations about the personalities and characteristics of users of the brand and their own personal experiences with the brand. These brand associations are compared to individuals’ self-concept—which may be their actual self or ideal self. Where brand associations and self-image are congruent, brand–self connection is formed [17]. Most brand associations relate to aspects of socialisation, including social status and peer group affiliation. Brand–self connection is demonstrated by the ease and frequency with which brand–self related thoughts and feelings are brought to mind [16]. The resonance of a brand with individuals’ self is the ultimate ambition of marketing, reflecting an intense physiological bond with the brand and active brand loyalty, characterised by repeat purchases, and a sense of community, kinship or engagement with the brand and other users of the brand [18]. For example, this kinship may manifest in online brand or brand community interactions.

These different levels of brand affect, from positive brand associations through to strong brand attachments, are developed over time with repeat brand exposures. The frequency of exposures, as well as the persuasive power of promotions, determines the potential for higher levels of brand affect [5,19]. From a public policy perspective, limiting children’s exposure to brand messages for unhealthy foods and beverages, and the persuasive power of these promotions, could stifle the cascade of effects of marketing by suppressing brand awareness, associations and attachments.

The symbolic meaning of food and beverage brands as identifiers of consumer identities has been explored in only a small number of studies. In one qualitative study of 7–14 year old British children that explored preferences for lunchbox items, older children were more likely to prefer branded food items that conformed to their perception of what their friends would like and for brands that conveyed acceptance and popularity [20]. An Australian study with 10–16 year olds (*n* = 417) identified strong positive affect for certain food and beverage brands, perceiving some unhealthy food brands to have positive attributes, desirable user traits, and alignment to their own personality [21]. Branding has also been shown to influence taste perceptions. A US study with 3–5 year old children (*n* = 63) found that children preferred the taste of identical foods and drinks when they were in McDonald’s packaging compared to unbranded packaging [22]. More evidence is available from the alcohol control field, in which brand allegiance (having a favourite alcohol brand) was significantly associated with the likelihood of drinking and amount consumed [23]. Children’s ownership of alcohol branded merchandise has also been linked with current and future drinking in cross-sectional and longitudinal studies [24].

This study explored how Australian children perceive their favourite food and beverage brands, in terms of their attachment to brands in relation to their self. The association between brand attachments and food marketing exposures was assessed, using television viewing as a proxy indicator, to establish the extent to which marketing exposures may be linked to the development of brand–self attachments. Brand attachments were also compared against food-consumption behaviours to identify possible downstream impacts of brand affect on children’s diets.

## 2. Materials and Methods

### 2.1. Sampling

The study was approved by the University of Wollongong Human Research Ethics Committee (HE15/242). A cross-sectional survey was conducted with 8–12 year old children attending community events for a cancer charity across New South Wales (NSW), Australia from September 2015 to April 2016. These events are run across the year and in a wide range of communities, with about 40,000 people in NSW taking part each year. Twelve events were identified for the survey, representing a geographical spread across NSW. Based on an earlier study in which we tested a tool to detect associations between food brand awareness and commercial television exposure [25], we estimated that a sample size of *n* = 76 would be required to detect significant differences in brand awareness between higher and lower commercial television viewers, with a power of 80% and alpha of 0.05. A larger sample of at least *n* = 200 children was sought to allow for comparisons between demographic groups.

### 2.2. Measurement Instruments

#### 2.2.1. Development Process

Two rounds of formative interviews were undertaken to inform the development of the quantitative questionnaire. The first round used exploratory interviews with six pairs of children to explore how children describe their experiences with, and perceptions of, food brands. Half of the interviews were conducted with males and half with females. Half of the pairs were from higher socio-economic families and half from lower socio-economic backgrounds. The sample included a spread of ages. To ensure a relaxed environment in which children could openly talk about the topic, interviews were conducted in homes and children were known to each other. Interviews consisted of a series of tasks aimed to get children to think about food brands and their attitudes and attachments towards frequently advertised brands, and took approximately 30 min. The second round involved face-to-face cognitive pilot interviews with *n* = 10 children separately, to test understanding and interpretation of the quantitative interview guide developed from the first round of interviews. The interview guide was then finalised for use. The final questionnaire included questions on brand awareness and children’s attitudes and attachment to food and drink brands. The results for children’s brand attachments are presented in this paper.

#### 2.2.2. Measures for Determining Brand Attachment

In the interviews, children were asked to name their favourite food and drink brands, and these were used to answer the questions on brand attachment. In the exploratory interviews, children experienced difficulty in expressing their brand attachments beyond taste and their experience of the brand. To attempt to distil perceptions of the brand itself, as distinct from the food/drink product, we asked children to think about the brand in relation to branding on a T-shirt. Children were asked if they would prefer to wear a branded or a plain T-shirt, with the purpose of removing taste of the product from the decision, and explore the extent to which children felt comfortable “embodying” the brand. Children were then asked how much they would be willing to pay for their T-shirt of choice, as a measure of the perceived value of brands. Next, children were asked to indicate their agreement or disagreement with a series of statements (see Table 1) about their favourite food and drink brands, with reference to the strength and prominence of their brand attachments. These statements were adapted from earlier research in the marketing field [16] and tested for their appropriateness for children. Response categories ranged from ‘strongly agree’ to ‘strongly disagree’.

#### 2.2.3. Measures of Advertising Exposure and Consumption of Advertised Food Items

Detailed questions captured information on children’s commercial and non-commercial television viewing habits (minutes/day for weekdays and weekends, summed to give cumulative weekly television time for commercial and non-commercial channels), including frequency of watching television live or pre-recorded (with the possibility to skip through advertisements). Consumption of frequently advertised unhealthy foods (*n* = 7) and drinks (*n* = 4) was also assessed using short food frequency questions, with responses ranging from never to [times] per week [26]. The number of times per week that foods/drinks were consumed was summed across product categories to derive a weekly food/drink intake score.

#### 2.2.4. Other Data

Children’s age, sex, parents’ education level and postcode of residence were captured. Postcode was used to determine socio-economic status using the Australian Bureau of Statistics Socio-Economic Indexes for Areas (SEIFA) [27].

### 2.3. Procedure

For the formative interviews, an experienced social research company (www.curlyquestions.com) was contracted by the researchers for recruitment and fieldwork. This company had experience in undertaking commercial research and exploring consumer responses to brands. For the quantitative interviews, two trained interviewers visited each event, from a pool of 11 interviewers. All interviewers attended a half-day training session prior to fieldwork on delivering the structured closed-ended interview guide. At each event, an interview stand was set up to recruit children and the study was promoted through the use of flyers and public announcements. Parents were requested to be present at the time of the interview to facilitate children’s responses to questions about food and drink consumption and television viewing. Written parental consent and verbal child assent was obtained for all participants. Participating children were given a $10 iTunes voucher.

### 2.4. Analyses

Statistical analyses were conducted using SPSS for Windows version 25 (IBM Corp, Armonk, NY, USA). Because television viewing data did not meet normality assumptions, a Mann-Whitney U test was used to compare T-shirt preference by amount of commercial and non-commercial television viewing. Kruskal Wallis one-way analysis of variance was used to compare television viewing by willingness to pay for the branded T-shirt.

Responses to the statements about attachment to children’s favourite food and drink brands were assessed by exploratory factor analysis, using principal components extraction and varimax rotation. This was done to condense the brand attachment questions into discrete underlying dimensions that were then used in regression analyses. The Kaiser-Meyer Olkin (KMO) measure of sampling adequacy suggested that the sample was suitable for factor analysis for both the food brand and the drink brand data (food brand questions KMO = 0.714, drink brand questions KMO = 0.772). Likert scales were reverse coded, so that a higher score indicated a stronger brand attachment.

Linear regression analyses were performed to describe the association between food and drink brand attachment and commercial and non-commercial television viewing time, controlling for age and weekly food/drink intake of frequently advertised foods. Outputs included partial regression coefficients, representing the amount by which brand attachment changed when one of the independent variables increased by one unit and all the other independent variables were held constant. There was one plausible but extreme outlier for commercial television viewing time, with this child watching 32 h of commercial television per week (median = 5 h, interquartile range 2.6 to 9). Removing this case from the analyses resulted in a slightly attenuated beta-coefficient and *p* value for the relationship between commercial television viewing and food and drink brand attachment for children who were exposed to advertising on television, but the relationship remained significant. Examination of leverage statistics indicated an acceptable Cook’s distance for this case, therefore the case was maintained for the final analyses.

## 3. Results

### 3.1. Sample Characteristics

Twelve community events were attended, with between 5 and 30 children interviewed at each event (*n* = 282). Just over half of the sample was made up of girls, with a mean age of 10 years (standard deviation (S.D.) = 1.4) (Table 2) More than 90% of participants were from low and medium socio-economic status (S.E.S.) areas.

Of the children who reported watching commercial television (*n* = 272), the majority ‘always’ (39%) or ‘mostly’ (25%) watched this live or at the time of broadcast (*n* = 174). Of those who recorded any programs for viewing later, 14% (*n* = 14) ‘never’, ‘rarely or ‘sometimes’ skipped through the advertisement breaks. Together these two groups are the children who were classified as being ‘exposed’ to television food advertisements (*n* = 188).

### 3.2. Favourite Food and Drink Brands

All children gave a valid response when asked their favourite food brand, and *n* = 252 children gave a valid response for their favourite drink brand (30 children responded with a generic drink type, e.g., lemonade). Cadbury was the most frequently reported favourite food brand (18% of children; chocolate brand), followed by McDonald’s and Subway (both 9%; fast food brands) and Smiths chips (5%; savoury crisp brand). Fanta was the most common favourite drink brand (20%; soda brand) followed by Coke (16%; soda brand), Sprite (7%; soda brand), Schweppes (6%; soda brand), Solo (6%; soda brand) and Powerade (6%; isotonic drink brand).

### 3.3. Food and Drink Brand Attachments

Almost one-quarter of children (23%, *n* = 65) indicated that they would prefer to wear a T-shirt displaying their favourite food brand than a plain T-shirt. Of these children, 61% were willing to pay ‘a little’ (*n* = 32) or ‘a lot’ (*n* = 8) more for this branded shirt. There was no difference in the frequency of consumption of unhealthy food and drinks between children preferring the food-branded T-shirt (mean food/drink intake score 19.3 (S.D. 10.26)) and those preferring the plain T-shirt (18.1 (S.D. 10.03) (t (279) = −0.84, *p* = 0.4)). Nor was there any difference in the frequency of unhealthy food and drink consumption depending on children’s willingness to pay for the food-branded T-shirt (“willing to pay more” mean score = 18.1, “not willing to pay more” mean score = 21.1, t (63) = 1.14, *p* = 0.3).

A greater proportion of children (41%, *n* = 100) indicated that they would prefer to wear a T-shirt displaying their favourite drink brand over the plain T-shirt. Children who preferred the drink-branded T-shirt were more frequent consumers of unhealthy foods and drinks (mean food/drink intake score 20.2 (S.D. 10.58)) compared to those who preferred the plain T-shirt (17.5 (S.D. 9.65); (t (250) = −2.05, *p* = 0.04)). However, for those who preferred the branded T-shirt, frequency of consumption was not associated with willingness to pay more for the drink-branded T-shirt (“willing to pay more” mean score = 19.8, “not willing to pay more” mean score = 20.8, t (98) = 0.461, *p* = 0.6).

Children who preferred the drink-branded T-shirt watched more commercial television than those who preferred the plain T-shirt (mean rank 140 vs. 118; *u* = 6279.5, *p* = 0.016). There was a similar, but non-significant association between higher commercial television viewing and preference for the food-branded T-shirt (mean rank 155 vs. 137; *u* = 6141.5, *p* = 0.1). Commercial television viewing time was significantly associated with willingness to pay more for the drink-branded T-shirt (mean ranks: willing to pay a lot more = 180; willing to pay a little more = 157; not willing to pay more = 154; prefer unbranded = 132; χ^2^(3) = 8.10; 0.04) (Figure 1). Again, there was a similar but non-significant trend in the distribution between the amount of commercial television viewing and willingness to pay for the food-branded T-shirt (mean ranks: willing to pay a lot more = 188; willing to pay a little more = 159; not willing to pay more = 141; prefer unbranded = 137; χ^2^(3) = 4.64; 0.20).

There was no association between preference for the branded or unbranded T-shirt by non-commercial television viewing time for either the food (*u* = 6824, *p* = 0.7) or drink brands (*u* = 6872, *p* = 0.2). Nor was there any association between willingness to pay and non-commercial television viewing time for either the food-branded T-shirt (χ^2^(3) = 0.47; 0.9) or drink-branded T-shirt (χ^2^(3) = 2.42; 0.5).

Most children agreed with the statement that they would like to consume their favourite food or drink brand soon (Table 1). The majority of children also agreed that the brand/s made them feel good, were popular amongst their peers, and were “just right for a person like [them]”.

In the exploratory factor analysis of the brand attachment survey items, two components emerged which explained approximately half of the variance for the whole set of variables. These were labelled as: (1) brand–self connection; and (2) brand socialisation. The two factors were based on inspection of the scree plot and eigenvalues (Table 3).

For each child, the mean of the Likert scale scores across questions within each factor were calculated, for food and drink brands separately. The higher the score for each factor, the more that children agreed with the brand statements, and therefore the stronger their brand–self connection or brand socialisation. In univariate linear regression models, children’s age was negatively associated with both food and drink brand–self connection, such that younger children reported stronger brand–self connection than older children. S.E.S. was not associated with brand self-connection or socialization.

Amount of commercial television viewing was only associated with food and drink brand–self connection for those children who reported being exposed to television advertisements. Those children who watched the most commercial television and who were exposed to the advertisements had the strongest brand connections. There was no association between food and drink brand–self connection and amount of commercial television viewing for those children who were not exposed to advertisements, nor was there any association with non-commercial television viewing. Both food and drink brand–self connection were also positively associated with the frequency of unhealthy food and drink consumption in univariate models, such that more frequent consumers had stronger brand–self connections.

In multivariate linear regression models, for those children who were exposed to television food advertising, the variables age, commercial television viewing and frequency of unhealthy food and drink consumption all had significant partial effects on drink brand–self connection and together explained 13% of the variation (F (3, 182) = 7.60, *p* = 0.001) (Table 4). For food brand–self connection, children’s age and amount of commercial television viewing had significant partial effects for those children who were exposed to advertisements. Together these covariates explained 11% of the variation (F (3, 162) = 8.20, *p* = 0.001).

There were no associations found between food or drink brand socialisation and age, commercial or non-commercial television viewing, or frequency of unhealthy food and drink consumption.

## 4. Discussion

The principal finding from this study was that food marketing exposure, as estimated based on commercial television viewing time, was positively associated with children’s attachments to their favourite food and drink brands. Children who watched a greater amount of commercial television had a stronger attachment to their favourite drink brands, demonstrated by their preference for the drink-branded T-shirt. The more commercial television that children watched, the more they were willing to pay for the drink-branded T-shirt and the stronger their agreement with statements about the drink brands’ connection to self. Commercial television viewing was also associated with food brand attachments, with a positive linear response between the amount of viewing and the strength of children’s attachment to their favourite food brand. Exposure to food advertising was responsible for these associations between television viewing and brand attachment. That is, the association was only found for children who were exposed to advertising content (watched television live or did not skip through advertisements in recordings). There was no association between the amount of non-commercial television viewing (which broadcasts no advertisements) and food or drink brand attachments.

Television food advertising was used as a proxy measure for children’s food marketing exposures more broadly. However, contemporary food marketing environments for children integrate promotional campaigns across media, social media and numerous other settings [28], such that children see commercial messages for unhealthy foods and beverages everywhere and all of the time. For example, innovative research from New Zealand, in which 11–13 year old children wore an automatic camera around their neck for four days, found that children had an average of 27 unhealthy food marketing exposures every day [29]. This exposure mostly excluded on-screen advertising, given technical issues in capturing screen content. Despite the evidence that children are exposed to marketing from multiple sources, the findings from our study, whereby there were differences between those exposed to television advertisements and those not exposed, suggests that television is a key media for creating brand attachment. However, while statistically significant, commercial television viewing contributed to only a small amount of the variation in children’s food and drink brand–self connections. The contribution that overall food marketing exposure makes to brand–self connection may be much larger if other marketing exposures beyond television were considered. Future studies should seek to better estimate children’s overall food marketing exposure and the relationship with children’s brand affect.

Much of the evidence in the marketing and psychology literature on brand–self connection has involved adults; however, from middle to late childhood (8 to 12 years) children have the requisite cognitive capabilities to develop brand–self connection. That is, they possess an understanding of themselves [30] and are able to conceptualise brands abstractly rather than considering these for more concrete attributes, such as taste [17]. In earlier surveys, children aged 10 to 16 years were able to assign connotative meaning to food brands and food brand users, perceiving some unhealthy food brands to have positive attributes and desirable user traits [21].

We found that brand–self connection was greatest at younger ages in our sample. This is at odds with earlier research, which found that while brand self-connections developed in middle childhood, this tended to be based on concrete associations related to their brand experiences [17]. In adolescents, brand self-connection had been found to be stronger and linked to their concept of brand–self personalities, user characteristics and peer group affiliations [17]. Arguments for the protection of children from the harmful impact of food marketing have centred on the vulnerability of younger children, given their relative cognitive immaturity and lack of understanding of the persuasive intent of advertising [31]. In our study, increasing advertisement understanding with age may have tempered children’s brand–self attachments. However, the discordance of this finding with earlier work highlights the need to explore food and drink brand–self connection in adolescents, perhaps using methods less reliant on self-report, which are subject to social desirability biases and demand characteristics. For example, physiological arousal measures, such as skin conductance, have been used in other studies with children aged 8–11 years to explore the implicit associations that young people make with their favourite food and beverage brands, comparing physiological responses to images of brands to their responses to images of family and friends [32]. Other neurological methods, such as functional magnetic resonance imaging (FMRI) have been used with 9–16 year olds, identifying that food logos activated brain regions known to be associated with motivation [33].

Compared to advertisement exposure, brand attachment itself was less associated with frequency of consumption of commonly advertised unhealthy foods and beverages. Frequency of unhealthy food and beverage consumption was only associated with a preference for the drink-branded T-shirt, but not the food-branded T-shirt, nor willingness to pay more for either branded shirt. At this age of mid- to late-childhood, parents are still the gatekeepers of children’s food intake. Children’s intentions to consume preferred products may be attenuated by parent responses, influencing the association between brand perceptions and children’s consumption behaviour. However, both food and drink brand self-connection were positively associated with frequency of unhealthy food and beverage consumption in the current study. Brand attachment may increase intake of branded products, as well as consumption of foods within the same category. Such ‘beyond-brand’ effects of television food advertising on food choice have been identified in experimental studies, in which children consumed more of specific food products and more food overall after exposure to food advertising [34]. We did not ask children about the frequency of consumption of the exact branded products for which they were questioned on their brand attachments, which may have weakened our ability to detect associations between brand connections and consumption. However, notably, almost all favourite food and drink brands reported were captured in the food and beverage categories used to estimate unhealthy consumption.

The findings of this study are aligned with earlier surveys, which identified a positive association between commercial television viewing and intake of unhealthy foods and drinks [35]. Again, this association only applied to those children who were actually exposed to advertisements embedded within programs. In accordance with hierarchy of effects models for food marketing, responses to marketing exposure progress through cognitive, to affective, to behavioural effects [4,36]. This study confirms that brand attachment is one intermediary response to marketing exposure, through which advertising exposure leads to increased intakes of promoted foods. Specifically, this study confirms that advertising exposure is linked to food and drink brand attachment and brand attachment is then, more weakly, linked with consumption of frequently advertised foods.

Notable limitations of the current study include the cross-sectional design, which limits inference regarding the direction of associations. For example, it is possible that children who have a stronger brand–self connection choose to consume more of the branded foods and also that consumption of the food promotes brand connections. As noted above, we also only captured exposure to television food advertising, while this comprises only one part of children’s food and drink brand interactions. While the study espouses the application of theoretical hierarchy of effects models to food marketing, by linking marketing exposure to brand affect and, separately, brand affect to consumption, no attempt was made to conduct mediational analyses to confirm the mediating role of brand affect between food marketing exposure and food consumption. Future studies with larger sample sizes are needed to test the relationships and pathways between cognitive, affective and behavioural outcomes of food marketing exposures.

## 5. Conclusions

Brand attachments appear to be an important part of the cascade of effects in food marketing and are an intermediary pathway through which marketing operates. This study identified an association between food marketing exposure, brand attachment and unhealthy food and drink consumption. This was despite the relative limitations of the measures used to capture marketing exposure and food and drink brand consumption, which were proxy indicators rather than precise measures. Despite this, the significant associations identified provide a framework for investigating the links more precisely in future research. This paper contributes methodologically to the field, given the substantial developmental work that went in to the preparation of the measurement tool for assessing brand attachments. As such, it provides practical ways of measuring brand attachment and affective responses to food marketing in children that are both feasible and acceptable. Given the association identified between food marketing exposure and the development of food and drink brand attachments, there is an obvious necessity to limit children’s food marketing exposures to attenuate such affective responses. Additionally, the persuasive appeals used within marketing communications serve to attract children and build brand image. Thus, policy responses to limit the impact of unhealthy food marketing on children should seek to limit children’s exposures to unhealthy food marketing and the persuasive power of such promotions. Globally, a number of countries have implemented regulatory responses to limit children’s exposure to, and the power of, unhealthy food marketing [37]; demonstrating that regulatory approaches are possible and can be effective in reducing food marketing exposure, power and impact (e.g., [38]).

## Figures and Tables

**Figure 1 ijerph-16-02358-f001:**
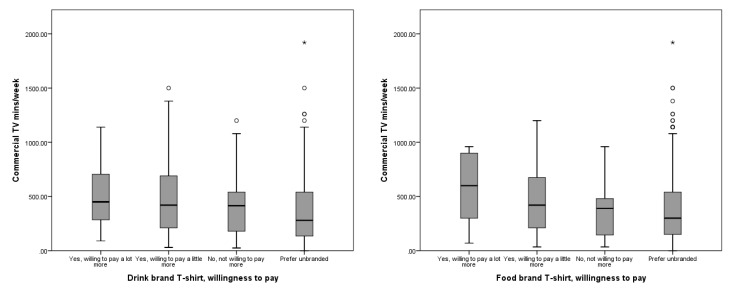
Distribution of commercial and non-commercial television viewing (mins/week) by willingness to pay for food- and drink-branded T-shirts.

**Table 1 ijerph-16-02358-t001:** Brand attachment for favourite food and drink brands.

Brand Perceptions	Food Brand n (%) Strongly Agree or Agree ^1^	Drink Brand n (%) Strongly Agree or Agree ^2^
I’d like to eat/drink [brand] sometime soon	231 (82)	196 (78)
[Brand] makes me feel good	212 (75)	170 (68)
[Brand] is popular amongst people my age	201 (71)	183 (73)
[Brand] is just right for a person like me	188 (67)	162 (64)
I think about [brand] regularly	45 (16)	61 (24)
I feel close to [brand], like the way I feel about a friend	64 (23)	44 (18)
Eating/drinking [brand] helps me fit in with other kids	49 (17)	48 (21)

^1^ Total *n* = 282; ^2^ Total *n* = 252.

**Table 2 ijerph-16-02358-t002:** Characteristics of the children’s sample.

	Children n (%)
Sex	
Female	164 (58)
Male	118 (42)
Age	
8 years	48 (17)
9 years	41 (15)
10 years	64 (23)
11 years	61 (22)
12 years	67 (23)
Relative SES of residential area ^1^	
Low SES	159 (57)
Medium SES	102 (36)
High SES	20 (7)
*Total*	*282 (100)*

^1^ Based on the Australian Bureau of Statistics Socio-Economic Indexes for Areas [27].

**Table 3 ijerph-16-02358-t003:** Component loadings for seven survey items for food and drink brands.

	Food Brand		Drink Brand	
	Factor 1 Brand–Self Connection	Factor 2 Brand Socialisation	Factor 1 Brand–Self Connection	Factor 2 Brand Socialisation
I feel close to [brand], like the way I feel about a friend	0.72	0.09	0.677	0.273
[Brand] is just right for a person like me	0.70	0.01	0.66	0.12
I think about [brand] regularly	0.65	0.11	0.67	0.27
[Brand] makes me feel good	0.50	0.38	0.72	−0.05
I’d like to eat/drink [brand] sometime soon	0.63	−0.08	0.71	0.00
[Brand] is popular amongst people my age	−0.24	0.77	−0.06	0.80
Eating/Drinking [Brand] Helps Me Fit in with Other Kids	0.28	0.69	0.29	0.70
Eigenvalue	2.29	1.17	2.68	1.06
% of total variance	31.58	17.87	34.93	18.45
Total variance		49.45		53.39

**Table 4 ijerph-16-02358-t004:** Summary of linear regression analyses for variables predicting food and drink brand–self connection.

	Food Brand–Self Connection (*n* = 280)						Drink Brand–Self Connection (*n* = 250)					
Variable	Exposed to TV ads			Not Exposed to TV ads			Exposed to TV ads			Not Exposed to TV ads		
	*B*	*SE B*	*β*	*B*	*SE B*	*β*	*B*	*SE B*	*β*	*B*	*SE B*	*β*
Age (years)	−0.12	0.04	−0.24 ***	−0.12	0.05	−0.24 *	−0.10	0.04	−0.18 *	−0.13	0.7	−0.20
Commercial TV viewing (hours)	0.02	0.01	0.18 *	−0.01	0.01	−0.07	0.03	0.01	0.21 **	−0.02	0.02	−0.12
Frequency food and drink consumption	0.01	0.01	0.09	0.01	0.01	0.12	0.01	0.01	0.18 *	0.02	0.01	0.23 *
*R* ^2^			0.11			0.08			0.13			0.09
*F*			7.60 ***			2.46			8.20 ***			2.78 *

* *p* < 0.05, ** *p* < 0.01, *** *p* < 0.001.

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
