# Peer review of "Children’s Exposure to Television Food Advertising Contributes to Strong Brand Attachments"

_ijerph, 2019, doi:10.3390/ijerph16132358_

Round 1
Reviewer 1 Report
Reviewer Comments
This study investigates the impact of television commercials for unhealthy foods on brand attachments and subsequent consumption of unhealthy foods among children. Findings from interviews and a survey indicate that length of exposure to television commercials reinforces food and beverage brand attachments, which in turn leads to consumption of unhealthy food products.
I enjoyed reading the paper; it is well written and very interesting. Here are a few comments that would further improve the manuscript.
Main Points
· In the ‘Measures for determining brand attachment sub-section’ (Materials and Methods), the authors note that in helping respondents delineate brand attachments from taste, they were asked to think in terms of a branded T shirt. While the researchers understandably did not want to prime their study sample about expressing favorite brands, it is unclear how this method helped distilled the information on brands. Also, how was brand attachment measured? Was the qualitative willingness to pay (WTP) measure intended to measure brand attachment? Additional explanation on these would be helpful in the Materials and Methods section. If Table 3 answers these questions, then it should be mentioned and indicated much earlier in the manuscript, again, in the Materials and Methods section.
· How was the frequency of television viewing/commercials captured quantitatively? Were these number of times(hours) per week a participant watched a particular commercial, as was the case for the food consumption frequency, or some other metric? Some extra detail in the text, specifically under sub-section iii of 2.2 in addition to a table/figure summarizing the statistics would be insightful.
· I suggest additional regressions to improve the discussion:
1. Even though only two variables loaded on the brand socialization factor, it would be interesting to see how the results change when brand socialization is treated as the dependent variable. In other words, would the same factors that influence brand-self connection also influence brand socialization?
2. As additional robustness, I suggest controlling for the residential area SES variable in the multivariate regressions. It would be interesting to see whether different socioeconomic zones predispose children to brand attachments – that would further enrich the discussion, and would be very policy relevant.
3. Why was preference for branded T shirt, or in variation, WTP for them not included in the multivariate regressions? Since they feature in the Discussion section, it would be useful to see how they influence brand attachment/socialization when other variables are controlled for.
· The discussion on brand-self connection in the “Discussion” section between lines 324 and 337 is long, and not necessarily based on the study’s findings. I suggest that the authors shorten this, and add the relevant literature to the ‘branding’ paragraphs in the introduction section.
· What could be an example of a policy response that limits children’s exposure to unhealthy food marketing? A more concrete example/suggestion in the Conclusion section would add some practical significance to the suggestion.
Minor Points
§ On line 69, self-identify should be self-identity
§ In the ‘Development process’ sub-section, what was the reasoning in using six pairs of children? Did each pair differ in some characteristics? An additional sentence expanding on that will be helpful for readers.
§ SES (Socio-economic status) from section 3.1 should be written in full before the abbreviation is used.
§ In lines 202 and 203, the 63% of participants who always or mostly watched TV commercials (39% + 24%) make ‘n’ approximately 171, not 174, i.e., if the total of that n=272. Please check.
§ In line 301, what is a “dose” response? It is a typo?
Author Response
Reviewer 1 - response
This study investigates the impact of television commercials for unhealthy foods on brand attachments and subsequent consumption of unhealthy foods among children. Findings from interviews and a survey indicate that length of exposure to television commercials reinforces food and beverage brand attachments, which in turn leads to consumption of unhealthy food products. I enjoyed reading the paper; it is well written and very interesting. Here are a few comments that would further improve the manuscript.
Thank you.
Main Points
In the ‘Measures for determining brand attachment sub-section’ (Materials and Methods), the authors note that in helping respondents delineate brand attachments from taste, they were asked to think in terms of a branded T shirt. While the researchers understandably did not want to prime their study sample about expressing favorite brands, it is unclear how this method helped distilled the information on brands. Also, how was brand attachment measured? Was the qualitative willingness to pay (WTP) measure intended to measure brand attachment? Additional explanation on these would be helpful in the Materials and Methods section. If Table 3 answers these questions, then it should be mentioned and indicated much earlier in the manuscript, again, in the Materials and Methods section.
Throughout the exploratory interviews, children tended to talk about brands in terms of the taste, or the healthiness (or lack of healthiness) of the products. In this way, the primary drivers for liking or disliking a brand related to the taste or experience of the product. However, while it was the case that children primarily spoke about brands in terms of the taste and healthiness or otherwise of the product, some children were able to identify other aspects of a brand that appealed to them. As the exploratory interviews progressed, a different line of questioning was adopted. Children were given a choice of a plain T-shirt, or a T-shirt showing a particular food/drink brand. Children were asked which shirt they would prefer to wear. The purpose of transferring the brand to an unrelated category was to remove the taste of the product from the decision, and explore the extent to which children felt comfortable “embodying” the brand. The follow-up question about whether they are prepared to pay more for their chosen shirt allowed some measure of the perceived value of brands. This explanation has now been included on page 4.
How was the frequency of television viewing/commercials captured quantitatively? Were these number of times(hours) per week a participant watched a particular commercial, as was the case for the food consumption frequency, or some other metric? Some extra detail in the text, specifically under sub-section iii of 2.2 in addition to a table/figure summarizing the statistics would be insightful.
Children were asked if they watched commercial and non-commercial television, separately. If so, they were then asked how many minutes or hours they watched of these channels on a typical weekday, Saturday and Sunday. These were added to give cumulative weekly time spent watching commercial and non-commercial television. This is further described on page 4.
I suggest additional regressions to improve the discussion:
1. Even though only two variables loaded on the brand socialization factor, it would be interesting to see how the results change when brand socialization is treated as the dependent variable. In other words, would the same factors that influence brand-self connection also influence brand socialization?
As described on page 8 of the original manuscript, when brand socialisation was inputted as the dependent variable in regression models, “there were no associations found between food or drink brand socialisation and age, commercial or non-commercial television viewing, or frequency of unhealthy food and drink consumption”.
2. As additional robustness, I suggest controlling for the residential area SES variable in the multivariate regressions. It would be interesting to see whether different socioeconomic zones predispose children to brand attachments – that would further enrich the discussion, and would be very policy relevant.
In univariate regression models, SES was not significantly associated with any of the dependent variables of food or drink brand self-connection or socialisation. As such, SES was not included in the final regression models. This is now indicated on page 8.
3. Why was preference for branded T shirt, or in variation, WTP for them not included in the multivariate regressions? Since they feature in the Discussion section, it would be useful to see how they influence brand attachment/socialization when other variables are controlled for.
Responses to the branded T shirt questions were not combined with the other questions on brand attachment in the exploratory factor analyses as the format of these questions varied considerably. That is, the T shirt questions asked children to choose between a branded or plain T shirt and then if they would be willing to pay a lot more or a little more for their T shirt of choice. Other attachment questions had a Likert scale as response categories. The differences in the structure of the questions and responses rendered these unable to be combined.
Given that the T shirt questions and the questions on agreement/disagreement with statements about brand attachment were ostensibly aiming to measure similar attributes (i.e. the strength and prominence of brand attachments), it would appear inappropriate to control for the T shirt responses in the regression analyses which had the agreement/disagreement with statements about brand attachment as the dependent variable.
The discussion on brand-self connection in the “Discussion” section between lines 324 and 337 is long, and not necessarily based on the study’s findings. I suggest that the authors shorten this, and add the relevant literature to the ‘branding’ paragraphs in the introduction section.
This section has now been moved to the introduction on Page 2.
What could be an example of a policy response that limits children’s exposure to unhealthy food marketing? A more concrete example/suggestion in the Conclusion section would add some practical significance to the suggestion.
A final sentence has been added to the Conclusion to address this point.
Minor Points
§ On line 69, self-identify should be self-identity
Amended
§ In the ‘Development process’ sub-section, what was the reasoning in using six pairs of children? Did each pair differ in some characteristics? An additional sentence expanding on that will be helpful for readers.
In these exploratory interviews, half of the interviews were conducted with males, and half with females, and half of the pairs were from higher socio-economic families, and half from lower socio-economic backgrounds. The sample included a spread of ages. This is now indicated on page 3.
§ SES (Socio-economic status) from section 3.1 should be written in full before the abbreviation is used.
Amended
§ In lines 202 and 203, the 63% of participants who always or mostly watched TV commercials (39% + 24%) make ‘n’ approximately 171, not 174, i.e., if the total of that n=272. Please check.
Thanks for picking up this rounding error. It should have read 25% of children mostly watched TV at the time of broadcast (i.e. 64%, n=174).
§ In line 301, what is a “dose” response? It is a typo?
This wording has been changed to “positive linear response”.
Reviewer 2 Report
Very good and interesting article. The Authors conducted interesting research and obtained reliable results. They applied the correct research methods. Authors should supplement the literature review in the article. Congratulations.
Author Response
Reviewer 2 - response
Very good and interesting article. The Authors conducted interesting research and obtained reliable results. They applied the correct research methods. Authors should supplement the literature review in the article. Congratulations.
Thank you for these positive comments. As per the comments from Reviewer 1, we have moved some of the discussion on brand-self connection/attachment to the introduction.
Round 2
Reviewer 1 Report
I am satisfied with the authors' responses.